# The Role of Circulating Lycopene in Low-Grade Chronic Inflammation: A Systematic Review of the Literature

**DOI:** 10.3390/molecules25194378

**Published:** 2020-09-23

**Authors:** Hidde P. van Steenwijk, Aalt Bast, Alie de Boer

**Affiliations:** 1Campus Venlo, Food Claims Centre Venlo, Faculty of Science and Engineering, Maastricht University, 5911 BV Venlo, The Netherlands; a.deboer@maastrichtuniversity.nl; 2Campus Venlo, University College Venlo, Maastricht University, 5911 BV Venlo, The Netherlands; a.bast@maastrichtuniversity.nl; 3Department of Pharmacology & Toxicology, Medicine and Life Sciences, Faculty of Health, Maastricht University, 5911 BV Venlo, The Netherlands

**Keywords:** carotenoids, phytochemicals, bioactive, nutrition, antioxidant paradox

## Abstract

Background and aims: In recent years, it has become clear that low-grade chronic inflammation is involved in the onset and progression of many non-communicable diseases. Many studies have investigated the association between inflammation and lycopene, however, results have been inconsistent. This systematic review aims to determine the impact of circulating lycopene on inflammation and to investigate the effect of consuming tomato products and/or lycopene supplements on markers of inflammation. Methods: Eligible studies, published before March 2020, were identified from PubMed, EBSCOhost and ScienceDirect. Human studies published in English, that evaluated the effect of circulating lycopene in relation to inflammation biomarkers were screened and included. Studies assessing lycopene intake or general intake of carotenoids/antioxidants without measuring circulating lycopene, as well as those not reporting inflammation biomarkers as outcomes, were excluded. Results: Out of 80 publications identified and screened, 35 met the inclusion criteria. Results from 18 cross-sectional studies suggest that lycopene levels are adversely affected during inflammation and homeostatic imbalance. Most of the 17 included intervention studies reported increased circulating lycopene levels after tomato/lycopene supplementation, but almost no changes in inflammation biomarkers were observed. Conclusions: There is little evidence that increasing tomato intake or lycopene supplementation diminuates this inflammation. However, depletion of lycopene may be one of the first signs of low-grade inflammation. The available data thereby imply that it is beneficial to consume lycopene-rich foods occasionally to stay healthy and keep circulating lycopene at a basal level.

## 1. Introduction

The understanding of health has changed in recent years: in addition to medicine and pharmacology, there has been an increasing interest in lifestyle medicine in which nutrition plays a pivotal role [1]. In addition to conventional drug therapies, lifestyle adjustments, such as dietary changes, are also advised to reduce disease. Diets with a high proportion of fruits and vegetables seem to have a particularly positive effect on nutritional status as well as different non-communicable diseases, such as heart diseases, neurodegenerative diseases, and diabetes type II. As most non-communicable diseases are partially affected by inflammation, more research is being conducted on potential anti-inflammatory substances derived from fruits and vegetables [2,3,4,5,6].

### 1.1. Low-Grade Chronic Inflammation

Previous research has shown that the onset and progression of many non-communicable diseases, including heart diseases, neurodegenerative diseases, and diabetes type II, are (partly) related to, or affected by inflammation: low-grade chronic inflammation is central to many different symptoms from which patients suffer in these conditions. Chronic inflammation is believed to aggravate various mechanisms that reflect poor health, including elevated blood pressure, high blood sugar, excessive waist circumference, and abnormal cholesterol or triglyceride levels (the so-called “deadly quartet”) [7]. In normal homeostasis, the function of inflammation is to eliminate the initial cause of cell injury, dispose of necrotic cells and damaged tissue caused by both the injury and the inflammation, and to initiate tissue repair. This natural response, acute inflammation, is a critical survival mechanism used by all higher vertebrates [8]. However, if acute inflammation is not resolved, it can lead to chronic inflammation, which is not part of the body’s natural healing process and can constitute a damaging process. Damaged tissues release pro-inflammatory cytokines and other biological inflammatory mediators into the circulation, converting tissue-based low-grade inflammation into a systemic inflammatory condition. Moreover, autoimmune disorders and long-term exposure to irritants can also lead to a systemic inflammatory condition [8,9,10].

The inflammatory response is the coordinated activation of signaling pathways that regulate inflammatory mediator levels in resident tissue cells and inflammatory cells recruited from the blood. Although inflammatory response processes depend on the precise nature of the initial stimulus and its location in the body, for example, bacterial pathogens trigger Toll-like receptors (TLRs) and viral infections trigger type I interferons (IFN), they all share a common mechanism, which can be summarized as follows: (1) Cell surface pattern receptors recognize detrimental stimuli; (2) inflammatory pathways are activated; (3) inflammatory markers are released; and (4) inflammatory cells are recruited [9,11]. Inflammatory stimuli activate intracellular signaling pathways that subsequently activate the production of inflammatory mediators. Primary inflammatory stimuli, including microbial products and cytokines such as interleukin-1β (IL-1β), interleukin-6 (IL-6), and tumor necrosis factor-α (TNF-α), mediate inflammation through interaction with the TLRs, IL-1 receptor (IL-1R), IL-6 receptor (IL-6R), and the TNF receptor (TNFR). This receptor activation triggers important intracellular signaling pathways, including the mitogen-activated protein kinase (MAPK), nuclear factor kappa-B (NF-κB), NF-E2 p45-related factor 2 (Nrf2), and Janus kinase (JAK)- signal transducer, and activator of transcription (STAT) pathways [11]. In the state of low-grade chronic inflammation, a typical inflammatory stimulator or pathogen can no longer be determined, and inflammatory stimuli and pathways remain activated. Inflammatory stimuli, such as IL-6 and C-reactive protein (CRP), can then be used as biomarkers to measure inflammation [12].

Low grade inflammation is involved in the progression of many non-communicable diseases, but also seems to affect apparently healthy people as a consequence of smoking, stress, or alcohol consumption [8]. A wealth of epidemiological evidence indicates that overall health is strongly influenced by diets with a high proportion of fruits and vegetables [2,3,4,13,14]. Phytochemicals with anti-inflammatory activity present in fruits and vegetables are believed to be largely responsible for overall health. Therefore, new possibilities may exist in the reduction and prevention of non-communicable diseases by increasing the intake of anti-inflammatory food (ingredients) in both healthy and diseased individuals [15,16,17].

### 1.2. Lycopene

One group of nutritional compounds that has been suggested to elicit anti-inflammatory effects are carotenoids. As carotenoids are pigments in photosynthetic tissue, they are ubiquitous in leafy green vegetables. In non-photosynthetic tissue, carotenoids are responsible for the characteristic coloration of fruits such as red tomatoes, orange carrots, and red flesh in watermelon [18,19]. Of all carotenoids, a substantial amount of research has been conducted on the acyclic lycopene, present in e.g.; tomatoes.

#### 1.2.1. Physicochemical Properties of Lycopene

Lycopene has a chemical formula of C40H56 and like all carotenoids, is a tetraterpene; assembled from eight isoprene units that are solely composed of hydrogen and carbon [20]. Lycopene is an acyclic isomer of β-carotene, however, unlike β-carotene lycopene lacks the β-ionic ring structure. Therefore, it lacks provitamin A activity [20,21]. However, lycopene is one of the most potent antioxidants, with a singlet-oxygen-quenching ability twice as high as that of β-carotene and ten times higher than that of α-tocopherol (Vitamin E) [22]. Lycopene is a highly unsaturated, open-chain hydrocarbon containing eleven conjugated and two non-conjugated double bonds arranged in a linear array. The double bonds in lycopene can undergo isomerization from *trans* to *cis* isomers by thermal energy, chemical reactions, and light [20,21]. The all-*trans* isomeric form is primarily present in nature, followed by the 5-*cis*, 9-*cis*, 13-*cis*, and 15-*cis* isomeric forms. Several methods for analysis of circulating lycopene are described. Methods differ in that (i) either plasma or serum lycopene is measured, (ii) multiple isomers, *trans*-lycopene or total lycopene are measured, (iii) circulating lycopene is adjusted for total cholesterol. The correction for total cholesterol has been made in more recent intervention studies because there is a risk of carotenoid status being misinterpreted in subjects on cholesterol-lowering therapy if they rely on crude serum or plasma levels.

#### 1.2.2. Lycopene Kinetics after Oral Administration: Absorption, Distribution, Metabolism, Excretion

Absorption of lycopene is similar to that of other lipid soluble compounds. Ingested lycopene is incorporated into dietary lipid micelles and absorbed across the gastrointestinal tract via passive diffusion into the intestinal mucosal lining. Then they are incorporated into chylomicrons and released into the lymphatic system for transport to the liver. Lycopene is transported by lipoproteins in the blood for distribution to the different organs [23]. Because of its lipophilic nature, the primary carrier of lycopene is LDL and not HDL [24]. Generally, 10–30% of dietary lycopene is absorbed with the remainder being excreted. The bioavailability of lycopene is greater from tomato paste than from fresh tomatoes. The increased absorption of lycopene from processed products is attributed to the presence of *cis* isomeric forms [25]. The absorption of lycopene in humans is influenced by several biological and lifestyle factors including gender, age, body mass index and composition, hormonal status, blood lipids concentrations, alcohol consumption, smoking, and the presence of other carotenoids in the consumed products [20]. When lycopene is administered as the all-*trans* isomer it rapidly isomerizes to a mixture containing more than 50% *cis*-isomers during absorption in the bloodstream and tissues. Moreover, a study showed that administration of all-*trans* lycopene in tomato sauce to human subjects for three weeks resulted in 77.3% *cis* isomers in prostate tissue and thus only 22.7% all-*trans* lycopene [26]. Liver, seminal vesicles, and prostate tissue are the primary sites of lycopene accumulation in humans [27]. Recent studies indicate that the accumulation in these sites may be due to the involvement of an active process for the uptake of carotenoids via the scavenger receptor class B type 1 protein (SR-B1) transporter, in addition to passive diffusion [28]. The full metabolic routes of lycopene in humans is still unclear. Only a few metabolites, such as 5,6-dihydroxy-5,6-dihydro-lycopene, have been detected in human plasma. It is suggested that lycopene may undergo in vivo oxidation to form epoxides which then may be converted to the polar 5,6-dihydoxy-5,6-dihydro-lycopene through metabolic reduction [29].

#### 1.2.3. Mechanism of Action (In Vitro)

Lycopene has been shown to inhibit the binding abilities of NF-κB and stimulatory protein-1 (SP1), and decreased expression of insulin-like growth factor-1 receptor (IGF-1R) and intracellular ROS concentrations in human SK-Hep-1 cells [30]. Recently, Fenni et al. [31] confirmed the potential involvement of lycopene in decreasing the binding abilities of NF-κB. They demonstrated the ability of lycopene supplementation to inhibit high-fat diet-induced obesity, inflammatory response, and associated metabolic disorder in mice. They evaluated the effect of lycopene on the phosphorylation of p65 and IκB, which are involved as modulators in the NF-κB pathway. Lycopene was able to strongly reduce phosphorylation of p65 and IκB, resulting in the deactivation of the NF-κB pathway, that previously was induced by the consumption of a high-fat diet. This effect can thus be seen as the induction of an anti-inflammatory effect. These results have also been observed in SW480 human colorectal cancer cells, where lycopene restrained NF-κB and JNK activation, resulting in a suppression of TNF-α, IL-1β, IL-6, COX-2, and iNOS expression. However, relatively high concentrations of lycopene were used (10–30 μM) compared to usual detectable plasma levels (1–2 μM) [32].

While *in vitro* and animal studies show promise for the potential health effects of lycopene, the relationship between lycopene and low-grade chronic inflammation in itself has so far been inconclusive in humans. Various systematic reviews have already been conducted on lycopene and how it affects different diseases and their symptoms, such as prostate and bladder cancer, Cardiovascular risk and metabolic syndrome [33,34,35,36]. The cross-sectional and intervention studies assessed in these reviews were often inconclusive, and the inconsistency among studies and the type of lycopene tested makes comparison difficult. The different lycopene measurements (self-reported FFQ, measurement of product, circulating lycopene) are a possible reason for the inconsistent results. Circulating measures are preferred for assessing relations, because self-reported measures of lycopene intake are subject to recall bias or memory error and intake measurements do not provide insight in the absorption, distribution, metabolism, and excretion of lycopene in the body. For in vivo studies, however, it is necessary to not just focus on lycopene intake but to actually measure the circulating lycopene concentrations in plasma or serum, in order to understand the health effects on humans [21]. C-reactive protein (CRP) and interleukin-6 (IL-6) are most commonly used to measure inflammation, but some studies have reviewed other inflammatory biomarkers (hyaluronic acid (HA), malondialdehyde (MDA), adiponectin, monocyte chemoattractant protein 1 (MCP-1), thiobarbituric acid reactive substances (TBARS), serum amyloid A (SAA), tumor necrosis factor-α (TNF-α), interleukin-1β (IL-1β)). These will also be included in this study [37,38]. As such, this is the first systematic review of the literature to investigate the relationship between circulating lycopene and inflammation.

## 2. Methods

### 2.1. Literature Search

This systematic review was conducted following the Cochrane and the Centre for Reviews and Dissemination guidelines on systematic reviews and is reported according to PRISMA guidelines [39,40]. This systematic review of the literature was conducted to investigate the relationship between circulating lycopene and inflammation in order to understand the health effects of lycopene in humans. To identify relevant human studies in which the relationship between lycopene and inflammatory markers was measured, a systematic search was conducted in Pubmed, EBSCOhost and ScienceDirect as databases. Using the Boolean search terms “serum lycopene” and “inflammation,” articles published in peer-reviewed journals in the English language were flagged for further review. As this review focuses on the effect of lycopene intake on inflammation in vivo, only human studies were considered for inclusion. The date of publication did not serve as an exclusion criterion. The search was conducted in duplicate by the first and last author and all potentially relevant publications up to March 2020 were included in the search.

The following search terms yielded 42 articles in Pubmed and 13 articles in EBSCOhost: ((“serum”[MeSH Terms] OR “serum”[All Fields]) AND (“lycopene”[MeSH Terms] OR “lycopene”[All Fields]) AND (“inflammation”[MeSH Terms] OR “inflammation”[All Fields])).

In ScienceDirect, the search terms “lycopene” and “inflammation” in “Find articles with these terms” and in “Title, abstract or author-specified keywords,” 79 articles published in English from peer-reviewed journals were flagged for further review in ScienceDirect. After removing duplicates (62), a total of 72 articles were therefore screened for inclusion in this systematic review. Additionally, the reference list of each included article was examined to identify any additional studies for inclusion that might not have appeared in the search results. This led to the identification of eight more articles for further review.

### 2.2. Application of Inclusion/Exclusion Criteria

As this systematic review focusses on circulating lycopene, only studies that report serum or plasma lycopene levels as independent measures and their relation to inflammation (inflammatory biomarkers) were considered for inclusion in this review. Although human studies were selected in the search terms, the search still identified a few in vitro and animal studies, which were subsequently excluded (3). Only full text, original human research studies were included. This led to the exclusion of two additional studies that presented research only as an abstract or in the form of a presentation.

Various studies were seen to report merely the intake of lycopene and its relationship with inflammation biomarkers, whereas the main interest of this study is to identify actual levels of lycopene in plasma/serum detected after/following consumption. Therefore, when analyzing full text versions of all studies, studies were excluded from this assessment based on the following exclusion criteria: (i) Assessment of lycopene intake or general intake of carotenoids/antioxidants without measuring circulating lycopene; and (ii) no inflammation biomarkers reported as outcomes. In total, the screening and eligibility process of this literature search led to the identification of 35 studies that were included and subjected to critical analysis (Figure 1).

### 2.3. Data Extraction and Analysis

To assess the quality and to minimize the risk of reporting bias, each author independently analyzed the articles. The authors discussed the extracted data and thoroughly considered differing interpretations before establishing consensus. Extracted information included: study design, randomization, duration and length of follow-up, methods of analysis, participants characteristics (population, settings of intervention, baseline characteristics), outcome measures (biomarkers), intervention details (i.e.; tomato or lycopene), and conclusions. Brief descriptions and summaries of results for the final articles included in this review are presented in Table 1: Cross-sectional studies assessing the relation between circulating lycopene and inflammation; and Table 2: Intervention studies assessing the influence of lycopene on inflammation.

## 3. Results

### 3.1. Study Characteristics

Of the 80 articles identified, screened, and considered for systematic review, 35 articles met the inclusion criteria for critical analysis. All papers were published between 1996 and 2018. Eighteen of the 35 studies used a cross-sectional study design and the remaining 17 were intervention trials. Studies varied widely not only in design and lycopene measurements, but also in the assessment of inflammation biomarkers. While some studies have identified multiple outcome measures, only the measurements of circulating lycopene and inflammatory biomarkers are highlighted in this review. Furthermore, studies that examined possible correlations or made conclusions regarding the relationship between circulating lycopene and inflammatory biomarkers, were also included in this systematic review. In various studies, the reported sample size for the outcomes of interest differed from the total number of participants. As such, these unique sample sizes have been reported with the corresponding measurement (Table 1 and Table 2).

### 3.2. Cross-Sectional Studies

As shown in Table 1 and Table 2, the discussion of the 35 articles is separated by study design. As displayed in Table 1, the results reported in the 18 included cross-sectional studies reviewed were grouped according to the following categories: type of lycopene measurement, assessment of inflammation biomarkers, and conclusions drawn from the study.

Five of these studies classified participants based on CRP concentrations or lycopene levels [41,43,45,51,53]. A study by Mazidi et al. [41] divided participants in quartiles depending on CRP concentrations and concluded that a higher lycopene level for each µmol/l correlated with 0.067 mg/dl lower CRP levels. Kritchevsky et al. [51] divided participants in tertiles depending on CRP levels and concluded that participants in the higher tertile CRP had significantly lower circulating lycopene levels. Furthermore, Boosalis et al. [53] divided elderly women (77–99 years) into two groups, based on either normal or elevated CRP levels, and showed that the presence of elevated CRP resulted in a significant decrease of lycopene concentrations. In addition, Kim et al. [43] divided healthy women (31–75 years) into tertiles according to serum lycopene concentrations, and reported that subjects in the highest tertile showed significantly lower CRP levels compared to those individuals in the lowest tertile. On the contrary, this association was not found in a study [45] in which young adults were divided into quartiles depending on the lycopene concentrations.

Thirteen other studies assessed the relationship between circulating lycopene and inflammation in healthy participants or patients. These studies report lower circulating lycopene concentrations and higher inflammation biomarker levels in patients with colorectal adenocarcinoma [42,55], carotid artery disease [44], stable angina pectoris [49,57], ischemic stroke [56], chronic hepatitis C [50], gastrointestinal cancer [52], benign prostate hyperplasia, localized and metastatic prostate cancer [54,55] and breast cancer [55] compared to healthy controls [48]. In addition, this relationship was also observed in critically ill patients [58], elderly disabled women [46], and people exposed to Schistosoma [47].

In general, these results suggest that lycopene levels are adversely affected during inflammation and homeostatic imbalance. These cross-sectional data do not clarify the biological relationship between lycopene and inflammation biomarkers. However, they do indicate the extent to which lycopene is associated with inflammation. They also indicate that the depletion of lycopene may be, in part, the first signs of low-grade inflammation.

### 3.3. Intervention Studies

As displayed in Table 2, the results reported in the 17 included intervention studies reviewed were grouped according to the following categories: type of lycopene measurement, assessment of inflammation biomarkers, type of intervention and conclusions drawn from the study.

Each of the included 17 intervention studies assessed lycopene levels and inflammatory biomarkers pre- and postintervention. All studies, except one, reported increased circulating lycopene levels following tomato/lycopene supplementation. In the exceptional study, supplementation with Lactolycopene capsules (supplements with lycopene entrapped with whey proteins) did not lead to a significant increase, but supplementation with Lycosome GA capsules (supplements with microencapsulated lycopene) did [63]. In a second study, supplementation with a combination of lycopene and rosuvastatin also did not significantly change plasma lycopene levels [70].

In ten intervention studies, biomarkers of inflammation were not reported to change after tomato/lycopene supplementation [59,62,64,66,67,68,70,73,74,75]. On the contrary, in a study by Li et al. [60], tomato juice supplementation led to a decrease of inflammatory adipokine MCP-1, and an increase in anti-inflammatory adiponectin levels in healthy Taiwanese females (20–30 years). Additionally, Biddle et al. [61] reported that tomato juice supplementation significantly decreased CRP levels in female heart failure patients, but not in male patients. Conversely, a decrease in hs-CRP was observed in healthy men following high lycopene (15 mg/day) supplementation [65] and after a single dose of tomato sauce (sofrito) [72].

Petyaev et al. [63] investigated the effect of supplementation with Lactolycopene or Lycosome GA capsules in patients with coronary artery disease. Serum lycopene levels of participants receiving Lactolycopene did not increase and CRP and MDA levels did not change after one month of supplementation. Nevertheless, in the group that received Lycosome GA capsules, circulating lycopene increased after one month, but only MDA was significantly reduced. In addition, opposite results were observed in healthy subjects in a study by Jacob et al. [69] in which CRP levels decreased following tomato juice supplementation, but IL-1β, TNF-α, and MDA levels remained stable.

Four of the seventeen selected studies conducted intervention studies in moderately overweight or obese individuals. Biomarkers for inflammation are often elevated in obese individuals compared to healthy individuals. One study [66] also used a healthy control group and concluded that pre-intervention CRP and IL-6 levels were significantly higher in obesity versus controls. All four concluded that markers of inflammation were not altered by lycopene, despite the significant increase in circulating lycopene after supplementation [62,66,67,75].

Four of the selected intervention studies investigated the effect of lycopene supplementation on inflammatory markers in patients with Cardiovasc. diseases [61,63,64,73]. These studies did not show consistent results. In one study, only MDA decreased [63]. In the next study, only CRP decreased in women (not in men) [61]. In the other two studies no alterations in inflammation biomarkers were observed after supplementation with lycopene [64,73]. The latter results were also observed in a study conducted in patients with type 2 diabetes, in which plasma lycopene levels increased nearly three-fold (*p* = 0.001), but no significant decreases in plasma levels of CRP were observed [68].

Six included intervention studies evaluated possible associations between lycopene/tomato supplementation and inflammation in healthy participants. In three of these studies, markers of inflammation did not change after supplementation, although circulating lycopene had increased by about 50 percent [59,64,69]. However, in another study the lycopene concentration also increased 1.5 times, and a significant decrease in hs-CRP was observed [65]. Furthermore, Hurtado-Barroso et al. [72] observed a three-fold increase in circulating lycopene and a significant decrease in CRP after a single dose of tomato sauce (sofrito). It is worth mentioning that CRP values in both studies were already below standard values before the start of the intervention. Li et al. [60] demonstrated that tomato juice supplementation led to a decrease of inflammatory adipokine MCP-1, and an increase in anti-inflammatory adiponectin levels in healthy young Taiwanese females. Compared to the other studies in which no or minor effects were seen on CRP, MCP-1 and adiponectin may be more sensitive biomarkers and therefore more suitable for studying inflammation in healthy individuals.

Rydén et al. [71] investigated the effect of simvastatin therapy on plasma lycopene levels and inflammatory markers in middle-aged men with mild to moderate hypercholesterolemia. Lycopene levels per total cholesterol (expressed as lycopene/total cholesterol) were significantly increased by simvastatin treatment. The findings may indicate that atherogenic lipoprotein particles have improved their antioxidant status through enrichment of carotenoids during simvastatin therapy.

Overall, most studies reported increased circulating lycopene levels after tomato/lycopene supplementation, but less than half of them observed alterations in inflammation biomarkers. In addition, two studies examined the effects of a low antioxidant diet in overweight women and asthmatic adults and observed a decrease in circulating lycopene and an increase in CRP [74,75]. Compared to supplementation, lycopene depletion appears to increase inflammation.

## 4. Discussion

This is, to our knowledge, the first systematic review to assess the correlation and causation between circulating lycopene (the bioavailable lycopene following consumption) and low-grade chronic inflammation. This review reveals that there is strong evidence indicating that lower circulating lycopene concentrations are related with higher inflammation biomarkers in patients with various diseases. In addition, this systematic review shows that there is little evidence that tomato intake or lycopene supplementation diminishes this inflammation.

In only one of the five studies in which CRP or lycopene levels were arranged into tertiles/quartiles, no association was found between circulating lycopene and CRP [41,43,45,51,53]. This could be attributable to the low CRP levels of the studied young adults (18–30); all mean CRP levels measured were between 0.99 and 1.11 mg/L [45]. On the contrary, the results from another study [43] showed a significant association and measured high-sensitivity CRP (hs-CRP) ranging from 0.80 and 1.27 mg/L. Moreover, when comparing the corresponding lycopene levels, it is striking that the values of Hozawa et al. [45] lie between 0.0242 and 0.0918 µmol/L, whereas most lycopene levels measured in all studies are between 0.1 and 1 µmol/L. It is therefore also possible that a non-reliable lycopene measurement has been carried out, so that no association could be found. The other three studies [41,51,53] did confirm the findings of Kim et al. [43], so there is strong evidence to suggest an association between circulating lycopene and CRP.

The eighteen studies evaluating the relationship between circulating lycopene and inflammation in healthy participants and patients gave similar results. These studies found lower circulating lycopene concentrations coincide with higher inflammation biomarkers in patients suffering from various diseases. These comparable results suggest that lycopene levels are adversely affected during inflammation and disturbed homeostasis. One possible explanation is that the development of oxidative stress during inflammation is responsible for the decreased lycopene levels. The prooxidant–antioxidant imbalance that ensues during oxidative stress may result in the increased utilization of endogenous and exogenous antioxidants, depleting circulating antioxidant concentrations. For that reason, any protective association that exists between serum lycopene and inflammation in patients may be attenuated [76,77,78,79]. Although the mechanisms underpinning reduced lycopene levels during inflammation are not fully elucidated, depletion of lycopene may be in part the first sign of low-grade inflammation.

Seventeen intervention studies were identified which better elucidate this carotenoid’s causal effect on inflammation and outcomes. Results from cross-sectional studies preclude the ability to ascribe causality because of both potential confounding and a lack of knowledge about the temporal relation between variables of interest. Most studies successfully increased lycopene levels through supplementation or tomato intake. In one study, supplementation with Lactolycopene capsules did not significantly increase lycopene levels. The authors emphasized the importance of proper supplement development, as another supplement increased circulating lycopene. In addition, supplementation with a combination of lycopene and rosuvastatin did not increase lycopene levels either [70]. The latter result could be explained by another study, in which supplementation with simvastatin, a comparable statin, led to a decrease in circulating lycopene. However, lycopene levels per total cholesterol were significantly increased following simvastatin treatment. The observed change in carotenoid status during simvastatin treatment was mainly attributed to the decrease in cholesterol, emphasizing the importance of cholesterol adjustment for expressing carotenoid levels [71].

This review found that the effect of lycopene supplementation or tomato intake on inflammation is incongruent: no changes in inflammation biomarkers were observed in half of the studies, and in the other half not all results were in line. Inflammatory markers were not altered by lycopene in moderately overweight or obese people, despite the significant increase in circulating lycopene after supplementation [62,66,67]. Intervention studies in patients with Cardiovascular disease or type 2 diabetes also showed minimal reduction of inflammatory markers [61,63,64,68]. In some intervention studies, it was stated that the intervention period was too short to observe a decrease in inflammatory biomarkers in patients. However, previous research has shown that treatment with non-steroidal anti-inflammatory drugs (NSAIDs) for a short period (two weeks) may reduce inflammatory biomarkers in patients, so these inflammatory biomarkers are unlikely to take longer to decrease [80,81]. Likewise, the results of lycopene supplementation in healthy participants were also inconsistent. Only two studies observed a significant decrease in hs-CRP after high lycopene supplementation (15 mg/day) or tomato sauce (sofrito) intake [72], but no effects were found after low lycopene supplementation (6 mg/day) [65] nor 7 mg/day [64]. The hs-CRP test accurately measures low CRP levels to identify low but persistent inflammatory levels. Therefore, it is more suitable for studying low-grade chronic inflammation in healthy participants in further research. However, it is debatable whether such a significant reduction in CRP below the standard values of 1–3 mg/L is clinically relevant and shows an actual anti-inflammatory effect, as these low CRP values already demonstrate that there is hardly any inflammation present. The other studies evaluating CRP report no significant changes in CRP levels following lycopene intake, probably because of the already low basal value in healthy participants. In addition, it would be of interest to evaluate new, more sensitive biomarkers in subsequent studies, as MCP-1 and adiponectin prove to be suitable biomarkers to study inflammation in healthy subjects [60].

Two intervention studies investigated the potential beneficial effects of lycopene in its isolated form (supplement) and via a lycopene-rich diet. These particular studies showed that both methods were successful in increasing circulating lycopene, but not in changing inflammation biomarkers [62,67]. These results suggest that the form in which lycopene is administered is of less importance than the absorption per se. For example, the absorption of lycopene can be improved by method of preparation such as adding olive oil [82]. Current literature indicates that the incorporation of a functional food with the compound of interest could potentially enhance these protective properties through the provision of an intact food matrix. However, more research is needed to elucidate these speculations. The matrix may provide a synergistic environment to promote the bioactivity of phytonutrients. However, this matrix also presents a challenge, since the direct effects of lycopene cannot be separated from other bioactive compounds within the food [83,84].

### 4.1. Molecular Mechanisms of Action

The incongruent results observed between the cross-sectional and intervention studies may be attributed to the different mechanisms of action of lycopene. Many in vitro studies elucidated the protective properties of carotenoids. As free radical scavengers, carotenoids react with reactive oxygen species (ROS) by three distinct mechanisms: (i) radical addition/adduct formation, (ii) electron transfer, and (iii) allylic hydrogen abstraction [85]. However, it is difficult to extrapolate the results of such studies because processes in the human body are more complex. It is probable that a number of factors may serve to decrease the antioxidant effectiveness of carotenoids in vivo*,* making them ineffective against certain ROS [86]. Furthermore, recent findings have shown that the participation of phytochemicals in redox metabolism is far more complicated than simply scavenging free radicals and avoiding oxidation of molecules. The cellular redox homeostasis is sustained by an overall and well-adjusted network of subcellular redox circuits that oscillate constantly depending on nutrients and energy supplies, genetic and epigenetic codes, and interactions with the external environment [87].

Barros et al. [87] suggest the hypothetical existence of the l NAD(*p*) +/NAD(*p*)H-responsive redox switch of eukaryotic cells that triggers distinct phenotypic fates depending upon cellular redox balance. This theory may explain the reduced lycopene levels in impaired situations as well as the paradoxical phenomenon where depletion of lycopene appears to increase inflammation, but lycopene supplementation does not decrease inflammation.

This theory suggests that an increase of the cellular antioxidant capacity (from dietary intake or generated endogenously) slides the antioxidant “seesaw” pivot point to the right, attenuating the magnitude of ROS/RNS production in the cell. However, an excessive antioxidant load in cells (sliding further to the right) could prevent beneficial processes mediated by the Nrf2−Keap1−EpRE system (Figure 2) [87,88]. Halliwell [89] describes the “antioxidant paradox” that supports this theory. The term “antioxidant paradox” is often used to refer to the observation that oxygen radicals and other ROS are implicated in several human diseases, but giving large doses of dietary antioxidants to human subjects has, in most studies, little or no preventative or therapeutic effect on inflammation. In addition, providing weak pro-oxidants to manipulate endogenous antioxidant levels may be a more useful approach for prevention of non-communicable diseases than is consumption of large doses of dietary antioxidants [89]. For example, it is well-known that physical activity increases the level of oxidative stress, but this appears to be beneficial to health. This same stimulus is in fact necessary to allow upregulation of endogenous antioxidant defenses, a phenomenon known as hormesis [90,91,92]. In addition to physical activity, various phytochemicals present in fruits and vegetables also can increase the level of oxidative stress and may exert health effects in other ways than lycopene.

Isothiocyanates from cruciferous vegetables can react directly with sulfhydryl residues of Keap1, causing the release of Nrf2. The ROS scavenging capacity of curcumin from turmeric is mainly attributed to its structure as a bis-α, β-unsaturated β-diketone of the two ferulic acid units, connected through a methylene group, which in addition can modify the thiol groups of Keap1, causing the release of Nrf2 [93]. The semi-synthetic flavonoid 7-mono-O-(β-hydroxyethyl)-rutoside (monoHER) acts as a double-edged sword in cells subjected to oxidative stress; the antioxidant offers direct protection by scavenging ROS and the oxidized monoHER adducts Keap1, causing the release of Nrf2 [94]. Alternatively, epigallocatechin gallate (from tea), cinnamaldehyde (from cinnamon), and resveratrol (from grapes) act on upstream kinases such as Akt, ERK, PI3K, PKC, and JNK causing the indirect release of Nrf2 from Keap1 [93]. Nrf2 is translocated to the nucleus and binds to the antioxidant response element (ARE) located in the promoters of genes coding for antioxidant and detoxifying enzymes. Nrf2/ARE-dependent genes code for several mediators of the antioxidant response, including glutathione S-transferases (GSTs), thioredoxin, NAD(*p*)H quinone oxidoreductase 1 (NQO-1), and heme oxygenase 1 (HO-1) [95]. Paradoxically, this reaction is considered weakly pro-oxidant [96]. The resulting oxidative stress supports the hormetic feedback and therefore leads to an endogenous increase in antioxidant defenses. It is possible that an excess of exogenous antioxidants may have detrimental effects on health by blocking the hormetic process [90,91]. In addition, various phytochemicals, such as the flavonoid quercetin from onions, can increase the endogenous antioxidant defenses in multiple ways. Similar to monoHER, oxidation products of quercetin are able to modify the thiol groups of Keap1, causing the release of Nrf2 and by up-regulation of Nrf2 through the regulation of both transcription and posttranscription sites and repression of Keap1 by affecting the posttranscription site [97,98].

### 4.2. Dietary Recommendations

A wealth of epidemiological evidence indicates that diets rich in plant products (grains, fruits and vegetables) contribute to overall health [2,3,4,99,100]. It is not clear whether this health-promoting effect is mainly attributable to the antioxidants in these plant products. However, the evidence suggests that antioxidants do play an important role in maintaining health: two studies included in this review examined a low antioxidant diet, during which a decrease in circulating lycopene and a subsequent increase in CRP was observed [74,75]. The available data thereby imply that it is beneficial to consume lycopene-rich foods occasionally to stay healthy and keep circulating lycopene at a basal level. It is preferable to consume lycopene through whole food sources such as tomatoes, rather than to ingest it through supplementation. This is because (i) lycopene is stable during preparation methods, (ii) other phytonutrients are also present in e.g.; tomatoes, (iii) the potential benefits of the food matrix, and (iv) the costs. But as this study shows, it is unlikely that taking additional lycopene will help restore health if inflammation is already present. Nevertheless, additional research is needed to determine evidence-based recommendations on the effect of long-term lycopene intake or supplementation and reduction of inflammation. In today’s society, antioxidants are considered healthy, partly because of results from in vitro studies. It is possible that the health effects of fruit and vegetables are due to the wide variety of bioactive substances in the food matrices and the synergy between the different mechanisms of action of these phytochemicals in the body. Nevertheless, the riddle of the “antioxidant paradox,” as described in Section 4.1, is yet to be fully deciphered. Phytochemicals in fruits and vegetables, both anti- and pro-inflammatory, appear to play a key role in this. In further research, it is important to consider the complexity of the endogenous antioxidant defense system [90]. Epidemiological evidence indicates that a multifactorial strategy of exercise, a healthy weight, no smoking, and a balanced diet that includes plenty of fruits, grains, and vegetables, is optimal to prevent low-grade chronic inflammation and maintain health overall [89].

### 4.3. Strengths and Limitations

This systematic review is one of the first studies that focusses on studying circulating lycopene measurements and its effect on inflammation, instead of merely the intake of lycopene. Self-reported measures of lycopene intake are subject to recall bias or memory errors, and do not provide insights into the body’s absorption, distribution, metabolism, and excretion of lycopene. Another strength of this study is that in this review, a clear distinction is made between effects reported in observational studies versus intervention studies. These reported effects are furthermore explained by incorporating details on the molecular mechanism of action of lycopene. The results of the cross-sectional studies are consistent with the findings of previous systematic reviews assessing the relationship between lycopene and vascular risk, metabolic syndrome, prostate, and bladder cancer [33,34,35,36].

However, suggestions made in these reviews about the effect of lycopene supplementation to reduce the risk of these diseases differ from the findings in this review, as this review highlights that there is little evidence that lycopene supplementation reduces inflammation.

Furthermore, this study is not without limitations. Overall, intervention studies were characterized by small sample sizes and short duration. In follow-up research, it would be of interest to investigate the effects of long-term lycopene supplementation on inflammation. Additionally, it is important to acknowledge that there might be publication bias in the intervention studies. It is known that positive results are published in scientific literature more often than negative or inconclusive ones. This study has also not been able to perform a meta-analysis to quantify the potential effects of lycopene because studies differed widely in lycopene and inflammatory biomarkers measurements. Lastly, even though all authors were involved in conducting the systematic search, setting inclusion criteria, and reviewing the inclusion of publications, selection bias may have affected the in- and exclusion of certain studies.

## 5. Conclusions

The available evidence indicates that lycopene levels are adversely affected during inflammation and homeostatic imbalance. Although the mechanisms underpinning these reduced lycopene levels are not fully elucidated, depletion of lycopene may be one of the first signs of low-grade inflammation. Even though supplementation with lycopene or an increased intake of tomatoes does result in an increase in circulating lycopene, there is little evidence that the lycopene increase also results in relieving this inflammation. This phenomenon, also known as the “antioxidant paradox,” limits the added value of lycopene supplementation in both patients and healthy individuals.

## Figures and Tables

**Figure 1 molecules-25-04378-f001:**
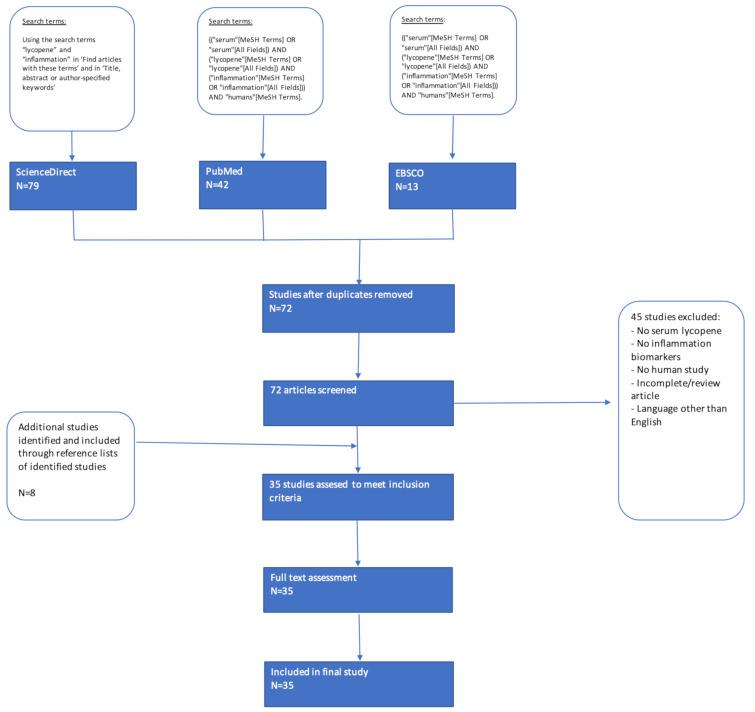
Flowchart of systematic search strategy.

**Figure 2 molecules-25-04378-f002:**
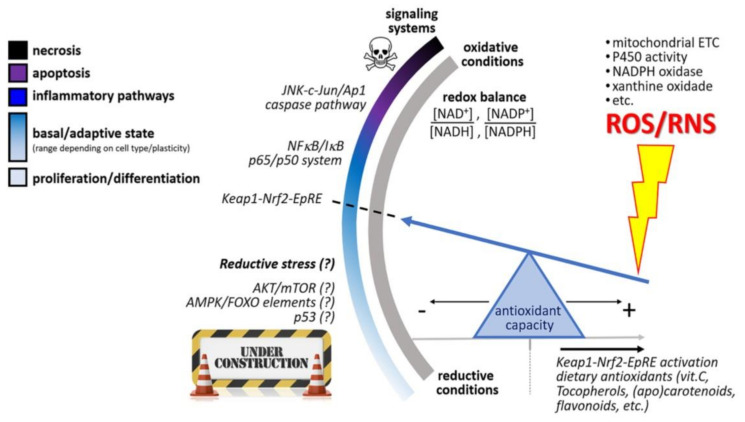
Hypothetical NAD(*p*) +/NAD(*p*)H-responsive redox switch of eukaryotic cells that triggers distinct phenotypic fates depending upon cellular redox balance. From a basal condition (optimum redox balance), the redox switch elicits inflammatory pathways, apoptosis, or necrosis, following increasing oxidative conditions, whereas unclear “reductive stress” mechanisms are triggered when NAD(*p*)H coenzymes prevail in cellular compartments. An increase of the cellular antioxidant capacity (from diet intake or generated endogenously) slides the antioxidant “seesaw” pivot point to the right, attenuating the magnitude of ROS/RNS production in the cell. However, an excessive antioxidant load in cells (sliding further to the right) could prevent beneficial processes mediated by the Nrf2−Keap1−EpRE system. This figure was adapted from reference [87].

**Table 1 molecules-25-04378-t001:** Cross-sectional studies assessing the relation between circulating lycopene and inflammation.

Study (Ref)	Study Population	Final *n*	Lycopene Measurement	Inflammation Biomarkers	Conclusions
*Mazidi* et al. [41]	Participants divided in quartiles depending on CRP and Fibrinogen	Q1: (*n* = 193) Q2: (*n* = 190) Q3: (*n* = 183) Q4: (*n* = 199)	Serum *trans*-Lycopene (μmol/L) Q1: 0.431 ± 0.007 Q2: 0.425 ± 0.007 Q3: 0.421 ± 0.005 Q4: 0.387 ± 0.009	CRP (mg/dL) Q1: 0.03 ± 0.01 Q2: 0.14 ± 0.04 Q3: 0.33 ± 0.07 Q4: 1.2 ± 0.89	A higher *trans*-lycopene level for each μmol/L correlated with 0.067 mg/dL lower CRP and 0.048 mg/dL Fibrinogen
Crespo-Sanjuán et al. [42]	Control subjects (*n* = 14) Patients with intestinal polyps (*n* = 39) Patients with colorectal adenocarcinoma (CRC) (*n* = 128)	Control (*n* = 14) Patients (*n* = 167)	Plasma Lycopene (μg/L) Control: 194.33 ± 66.17 Carc. in Situ: 138.57 ± 106.62 Cancer IV: 100.42 ± 71.20	Plasma CRP (mg/L) Control: 2.05 ± 2.33 Carc. in Situ: 13.93 ± 26.53 Cancer IV: 41.83 ± 62.01	Levels of lycopene were higher in the control group and low in the stage-IV group (*p* = 0.03), and were inversely correlated with CRP (*p* = 0.005, R = −0.215). We found a consistent relationship between high lycopene and absence of atherosclerosis (*p* = 0.002).
Kim et al. [43]	Healthy women (31–75 yrs) classified into tertiles according to serum lycopene concentration (*n* = 264)	T1 (*n* = 88) T2 (*n* = 88) T3 (*n* = 88)	Serum Lycopene (mmol/L) T1: 0.029 ± 0.000 T2: 0.039 ± 0.000 T3: 0.052 ± 0.001	hs-CRP (mg/dL) T1: 1.27 ± 0.24 T2: Data not shown T3: 0.80 ± 0.25	Subjects in T3 showed lower C-reactive protein (hs-CRP) (0.80 ± 0.25 mg/dL vs. 1.27 ± 0.24 mg/dL, *p* = 0.015), compared with those in T1.
Riccioni et al. [44]	Participants asymptomatic with respect to carotid artery disease divided over 3 groups based on Carotid intima-media thickness (*n* = 640)	C1 (*n* = 291) C2 (*n* = 232) C3 (*n* = 117)	Plasma Lycopene (μmol/L)C1: 0.82 ± 0.33C2: 0.33 ± 0.63C3: 0.34 ± 0.21	CRP (g/dL)C1: 2.90 ± 1.30 C2: 3.84 ± 1.75 C3: 4.86 ± 2.20	Elevated CIMT was significantly associated with having a low concentration of all antioxidants evaluated (vitamin A, vitamin E, lycopene, and b-carotene) and a higher concentration of inflammatory factors including serum uric acid, CRP, and fibrinogen.
Hozawa et al. [45]	Men and women in the Coronary Artery Risk Development in Young Adults study (18–30 years) divided in quartiles depending on Lycopene levels (*n* = 4580)	Q1: (*n* = 1144) Q2: (*n* = 1144) Q3: (*n* = 1144) Q4: (*n* = 1148)	Serum Lycopene (nmol/L) Q1: 24.2 Q2: 44.1 Q3: 62.0 Q4: 91.8	CRP (mg/L)Q1: 1.04Q2: 1.11Q3: 0.99Q4: 1.11	Serum total and individual carotenoids, with the exception of lycopene, were inversely associated with markers of inflammation
Walston et al. [46]	Subjects were disabled women aged >65 years (*n* = 619)	(*n* = 619)	Serum Lycopene (μmol/L) 0.56 ± 0.31	IL-6 (pg/mL) 5.51 ± 12.69	Persons with the highest levels of b-carotene, lycopene, lutein/zeaxanthin, b-cryptoxanthin, and retinol were also significantly less likely to be in the highest interleukin-6 tertile.
Eboumbou et al. [47]	Sudanese subjects exposed and not exposed to Schistosoma infection and French control subjects	Rural Sudan: (*n* = 35) Urban Sudan: (*n* = 27) French: (*n* = 34)	Serum Lycopene (μM)/Lycopene:B-carotene ratio RS: 0.21 (0.04)/1.10 US: 0.68 (0.10)/5.11 F: 1.10 (0.25)/4.52	Hyaluronic acid (HA)/Malondialdehyde (MDA) around 60 μg/L/200 nM	Drastic decrease of lycopene levels in the subjects exposed to schistosomiasis in comparison with non-exposed Sudanese and French control subjects
*van Herpen-Broekmans* et al. [48]	Healthy men and women (*n* = 379)	Men: (*n* = 178) Women: (*n* = 201) Total: (*n* = 379)	Serum Lycopene (μmol/L) Men: 0.35 ± 0.18 Women: 0.37 ± 0.18 Total: 0.36 ± 0.18	CRP (mg/L) Men: 0.9 (0.2–5.9) Women: 1.4 (0.2–7) Total: 1.1 (0.2–6.7)	An inverse relation between lycopene and CRP (−1.14 ± 0.54 per umol/l; *p* = 0.04) was found in men and not in women (0.50 ± 0.50 per umol/l; *p* = 0.32)
*Jonasson* et al. [49]	Men with stable angina and angiographically verified CAD and healthy controls (*n* = 113)	Patients: (*n* = 44) Controls: (*n* = 69)	Serum Lycopene (nmol/L) Patients: 177 (115–242) Controls: 298 (212–408)	CRP (mg/L) Patients: 2.30 (1.35–4.41) Controls: 1.22 (0.66–2.16)	Compared with controls, patients had signs of an enhanced inflammatory activity assessed by significantly increased levels of CRP. Patients also had significantly lower B-carotene and lycopene levels.
*Dhiraj* et al. [50]	Patients with Chronic Hepatitis C and controls (*n* = 42)	Patients: (*n* = 20) Controls: (*n* = 22)	Serum Lycopene (μg/dL) Patients: 6.2 ± 3 Controls: 59 ± 28	MDA (μM) Patients: 1.62 ± 0.57 Controls: 0.23 ± 0.15	Serum MDA levels were significantly higher in CHC patients compared with controls (1.62 ± 0.57 vs. 0.23 ± 0.15 μmol/L) Serum levels of lycopene were significantly decreased in CHC patients.
*Kritchevsky* et al. [51]	Nonsmoking participants aged 25–55 years (*n* = 4557) divided in tertiles depending on CRP levels	C1 (*n* = 3180) C2 (*n* = 924) C3 (*n* = 453)	Serum Lycopene (μmol/L) C1: 0.46 ± 0.004 C2: 0.45 ± 0.006 C3: 0.41 ± 0.010	CRP (mg/dL) C1: < 0.21 C2: 0.22–0.88 C3: >0.88–12.8	Lycopene is significantly lower in higher CRP tertile
*McMillan* et al. [52]	Healthy control subjects and patients with gastrointestinal cancer (*n* = 24)	Patients: (*n* = 12) Controls: (*n* = 12)	Plasma Lycopene (μmol/L) Patients: <0.02 ( <0.02–0.10) Controls: 0.37 (0.15–0.76)	CRP (mg/L) Patients: 91 (5–182) Controls: <5 ( <5–10)	The cancer group had significantly higher C-reactive protein concentrations (*p* < 0.001) and concentrations of lycopene were significantly lower (*p* < 0.001)
*Boosalis* et al. [53]	Catholic sisters (nuns) age 77–99 years (*n* = 85) divided in 2 groups depending on CRP levels	Elevated CRP: (*n* = 10) Normal CRP: (*n* = 75)	Plasma Lycopene (μg/dL) Elevated CRP: 9.0 ± 4.0 Normal CRP: 16.6 ± 10.6	Serum CRP (mg/dL) Elevated CRP: > 1.5 mg/dL Normal CRP: < 1.5 mg/dL	Results showed that the presence of elevated CRP resulted in a significant decrease of lycopene concentrations (*p* = 0.03)
*Almushatat* et al. [54]	Healthy subjects (C) Patients with benign prostate hyperplasia (B) Localized (L) Metastatic prostate cancer (M) (*n* = 112)	C: (*n* = 14) B: (*n* = 20) L: (*n* = 40) M: (*n* = 38)	Plasma Lycopene (μg/L) C: 127 (17–320) B: 128 (18–223) L: 83 (14–687) M: 42 ( <10–226)	MDA (μmol/L) C: 0.73 (0.50–1.40) B: 0.74 (0.35–1.48) L: 0.93 (0.47–2.93) M: 1.01 (0.44–4.67)	Prostate cancer patients had higher concentrations of malondialdehyde (*p* < 0.05) and lower circulating concentrations of lycopene (*p* < 0.001). There was a negative correlation between MDA concentrations and lycopene
*McMillan* et al. [55]	Healthy subjects (C) Breast cancer patients (B) Prostate (*p*)Colorectal (R) (*n* = 71)	C: (*n* = 30) B: (*n* = 15) *p*: (*n* = 15) R: (*n* = 11)	Plasma Lycopene (μg/100 mL) C: 18.0 (6.0–41.0) B: 1.8 ( <1.0–14.6) *p*: 6.7 (1.5–47.1) R: <1.0 ( <1.0–5.6)	CRP (mg/L) C: 2.0 (0.2–8.5) B: 3.9 (0.29–14.0) *p*: 8.0 (4.0–123) R: 70 (5.0–182)	Concentrations of CRP were higher and vitamin antioxidants lower in the cancer patients. In normal subjects and cancer patients, CRP concentrations were inversely correlated with circulating concentrations of lycopene.
*Chang* et al. [56]	Healthy controls (H) Ischemic stroke patients, small (S) or large artery (L) (*n* = 109)	H: (*n* = 41) S: (*n* = 35) L: (*n* = 33)	Plasma Lycopene (μmol/L) H: 0.13 ± 0.09 S: 0.10 ± 0.07 L: 0.09 ± 0.07	hs-CRP (mg/L) H: 1.6 ± 1.7 S: 6.0 ± 7.0 L: 8.4 ± 15.4	hs-CRP concentrations are significantly higher in patients with acute ischemic stroke than in healthy controls. Plasma lycopene, was inversely and significantly correlated with CRP.
*Chung* et al. [57]	Patients with stable angina (SA) or acute coronary syndrome (ACS) (*n* = 193)	SA: (*n* = 134) ACS: (*n* = 59)	Plasma Lycopene (μM) SA: 0.41 (0.25–0.65) ACS: 0.37 (0.26–0.58)	IL-6 (pg/mL) SA: 2.21 (1.45–3.03) ACS: 5.01 (2.68–9.36)	Only lutein + zeaxanthin was inversely correlated with IL-6 in SA patients at baseline
*Quasim* et al. [58]	Healthy controls (H) and critically-ill patients (C) (*n* = 67)	H: (*n* = 24) C: (*n* = 43)	Plasma Lycopene (μg/L) H: 189.0 (62.0–465.0) C: 15.5 ( <10.0–137.0)	CRP (mg/L) H: <5 C: 204 (6–345)	Systemic inflammatory response is associated with low carotenoid concentrations

**Table 2 molecules-25-04378-t002:** Intervention studies assessing the influence of lycopene supplementation on inflammation.

Study (Ref)	Study Population	Intervention	Final *n*	Lycopene Measurement	Inflammation Biomarkers	Conclusions
*Nieman* et al. [59]	Healthy runners (*n* = 20)	Lycopene capsule (5 mg/d) or placebo for 4 weeks	(*n* = 20)	Plasma Lycopene (ng/mL) Pre-supplement: around 500 Post-supplement: around 750	CRP (mg/L) Pre-supplement: 1.21 ± 1.2 Post-supplement: 1.28 ± 1.0	Plasma lycopene increased significantly in intervention group compared to placebo (*p* < 0.001). No alterations in post-exercise measures of oxidative stress and inflammation were found.
*Li* et al. [60]	Healthy young Taiwanese females (*n* = 25)	100% pure tomato juice, containing 11.6 mg of lycopene per 100 mL 280 mL/day for 56 days	(*n* = 25)	Serum Lycopene (μM) Pre-supplement: 0.72 ± 0.36 Post-supplement: 1.94 ± 0.74	Adiponectin (μg/mL) Pre-supplement: 11.5 ± 5.8 Post-supplement: 14.4 ± 5.2 MCP-1 (pg/mL)Pre-supplement: 126 ± 36 Post-supplement: 97.3 ± 17.9 TBARS (nM) Pre-supplement: 2.35 ± 1.11 Post-supplement: 1.84 ± 0.89	Tomato juice supplementation resulted in a decrease in levels of the inflammatory adipokine MCP-1, and an increase in levels of the anti-inflammatory adipokine adiponectin.
*Biddle* et al. [61]	Patients NYHA class II or III (*n* = 40)	V8 juice containing 29.4 mg of lycopene/day for 30 days	Control (*n* = 18) Intervention (*n* = 22)	Plasma Lycopene (μmol/L) Control, pre-supl: 0.56 Control, post-supl: 0.58 Intervention, pre-supl: 0.51 Intervention, post-supl: 0.76	Serum CRP (mg/L) Control, pre-supl: 4.8 ± 3.4 Control, post-supl: 4.5 ± 3.8 Intervention, pre-supl: 3.4 ± 3.1 Intervention, post-supl: 3.1 ± 2.8	C-reactive protein levels decreased significantly in the intervention group in women and but not in men (*p* = 0.04).
*McEneny* et al. [62]	Moderately overweight, middle-aged individuals (*n* = 54)	Control diet ( <10 mg lycopene/week)lycopene-rich diet (224–350 mg/week) lycopene supplement (70 mg/week)for 12 weeks	Control diet (*n* = 18) Lycopene diet (*n* = 18) Lycopene supl (*n* = 18)	Serum Lycopene (mmol/L) Baseline Control: 0.26 (0.03) Lycopene diet: 0.41 (0.04) Lycopene supl: 0.29 (0.03) Week 12 Control: 0.27 (0.03) Lycopene diet: 1.14 (0.05) Lycopene supl: 0.87 (0.06)	Serum Amyloid A (SAA) (μg/L) Baseline Control: 16,269 Lycopene diet: 15,566 Lycopene supl: 16,899 Week 12 Control: 18,882 Lycopene diet: 17,038 Lycopene supl: 12,070	Lycopene supplement tended to produce a greater response in reducing SAA concentrations and in influencing HDL’s function compared to the high-tomato diet.
*Petyaev* et al. [63]	Patients with coronary vascular disease (*n* = 142)	7 mg of lycopene/day for 1 month, two different lycopene supplements	Lactolycopene (L1) (*n* = 68) Lycosome GA (L2) (*n* = 74)	Serum Lycopene (ng/mg cholesterol) Baseline L1: 58.0L2: 55.0 Week 4 L1: 87.0 L2: 237.0	CRP (mg/L)/MDA (μM) Baseline L1: 6.0/141.0 L2: 6.8/154.0 Week 4 L1: 6.2/156.0 L2: 6.1/51.0	Lycopene supplementation had no impact on serum CRP level. Lactolycopene did not affect inflammatory markers by the end of the interventional period, whereas lycosome-formulated lycopene significantly reduced MDA
*Gajendragadkar* et al. [64]	Statin treated CVD patients and healthy controls (*n* = 72)	7 mg lycopene (1) or placebo (2)/day for 2 months Patients (*p*) and Healthy (H)	P1: (*n* = 24) P2: (*n* = 12) H1: (*n* = 24) H2: (*n* = 12)	Serum Lycopene (μg/L) Baseline/Day 56 P1: 146/275 P2: 128/178 H1: 170/267 H2: 182/160	hsCRP (mg/L)/IL-6 (pg/mL)/TNF-a (pg/mL) Baseline P1: 2.13/1.54/2.13 P2: 1.45/1.20/5.55 H1: 1.15/1.32/5.39 H2: 2.83/0.92/5.55 Day 56 P1: 2.37/1.51/2.37 P2: 1.68/0.92/5.65 H1: 1.87/1.02/4.92 H2: 1.65/0.84/5.32	hsCRP, IL-6 and TNF-a levels were unchanged for lycopene vs. placebo treatment groups in the CVD arm as well as the HV arm
*Kim* et al. [65]	Healthy men (*n* = 126)	Placebo (*p*) Low lycopene, 6 mg/d (L) High lycopene, 15 mg/d (H) For 8 weeks	*p*: (*n* = 38) L: (*n* = 41) H: (*n* = 37)	Serum Lycopene (μg/mL) Baseline/8 weeks*p*: 0.2/0.2 L: 0.2/0.26 H: 0.2/0.33	hsCRP (mg/dL)Baseline/8 weeks *p*: 1.14 ± 0.22/1.10 ± 0.27 L: 1.39 ± 0.33/1.40 ± 0.37 H: 1.25 ± 0.44/0.54 ± 0.10	A reduction in hs-CRP in the 15-mg lycopene/day group and the inverse correlation between changes in lycopene and changes in hs-CRP in this study, suggest that lycopene may play a role in inflammatory processes by interfering the action of cytokines.
*Markovits* et al. [66]	Obese patients (*p*) and healthy controls (C) (*n* = 16)	Patients received Lyc-o-mato, 30 mg/d for 4 weeks	*p*: (*n* = 8) C: (*n* = 8)	Serum Lycopene (μg/mL) C: 0.14 ± 0.07 *p*;baseline: 0.23 ± 0.22 *p*;supple: 1.15 ± 0.21	CRP (mg/L)/IL-6 (pg/mL)/TNF-a (pg/mL) Baseline C: 1.1/1.0/1.4 *p*: 6.5/3.6/1.4 Week 4 *p*; placebo: 5.5/3.5/1.4 *p*; supple: 5.6/4.7/1.5	CRP and IL-6 levels were significantly higher in obese vs. controls. Following lycopene treatment, a significant elevation of lycopene (1.15 vs. 0.23 μg/mL) (*p* < 0.001) occurred in the treatment vs. the placebo group. Markers of inflammation were not altered by lycopene.
*Thies* et al. [67]	Moderately overweight, disease-free, middle-aged adults (*n* = 225)	Control diet (C) High-tomato diet (H) Lycopene capsules (10 mg/d) (L) for 12 weeks	C: (*n* = 76) H: (*n* = 81) L: (*n* = 68)	Plasma Lycopene (μg/mL) Baseline/12 weeks C: 0.4/0.4 H: 0.4/1.1 L: 0.4/0.85	hsCRP (mg/L) Baseline/12 weeks C: 3.18/2.08 H: 1.51/1.37 L: 2.27/2.16 IL-6 (pg/L) Baseline/12 weeks C: 1.37/1.38 H: 1.21/1.15 L: 1.44/1.31	None of the inflammatory markers changed significantly after the dietary intervention. These data indicate that a relatively high daily consumption of tomato-based products (equivalent to 32–50 mg lycopene/d) or lycopene supplements (10 mg/d) is ineffective at reducing conventional CVD risk markers in moderately overweight, healthy, middle-aged individuals.
*Upritchard* et al. [68]	Patients with well-controlled type 2 diabetes aged <75 years (*n* = 57)	Placebo (C) Tomato juice 500 mL/d (T) for 4 weeks	C: (*n* = 13) T: (*n* = 15)	Plasma Lycopene (μmol/L) Baseline/4 weeks C: 0.31/0.28 T: 0.39/1.08	Plasma CRP (mg/L) Baseline/4 weeks C: 3.1/3.1 T: 3.8/4.1	Plasma lycopene levels increased nearly three-fold (*p* = 0.001) and no significant decreases in plasma levels of CRP
*Jacob* et al. [69]	Healthy subjects (*n* = 24)	2 weeks depletion followed by 2 weeks tomato juice 500 mL/d (41 mg/L lycopene, 90 mg/L Vitamin C) (L) or enriched with Vitamin C (870 mg/L) (LC)	T-2: baseline T0: after depl. T + 2: after inter. L: (*n* = 12) LC: (*n* = 12)	Plasma Lycopene (μmol/L) L/LC T-2: 0.72/0.71 T0: 0.42/0.34 T + 2: 1.05/0.91	L/LC CRP (ug/L) T-2: 336.2/349.5 T0: 315.6/319.2 T + 2: 262.3/247.1 IL-1 B (ng/L) T-2: 3.45/12.59 T0: 3.87/10.68 T + 2: 4.39/6.40 TNF-a (ng/L) T-2: 6.97/2.93 T0: 6.01/3.35 T + 2: 3.45/3.28 MDA (μmol/L)T-2: 0.55/0.60 T0: 0.54/0.56 T + 2: 0.53/0.50	The consumption of tomato juice led to a reduction of CRP in both groups. All other markers were affected to a lesser extent or remained unchanged.
*Williams* et al. [70]	COPD patients (*n* = 11)	Rosuvastatin (20 mg/day) for 4 weeks then a combination of rosuvastatin (20 mg/day), DHA and EPA (1.5 g/day) and lycopene (45 mg/day) for 8 weeks.	T1: baseline T2: rosuvastatin T3: lycopene	Plasma Lycopene (mg/L) T1: 0.30 (0.13–0.54) T2: 0.56 (0.14–0.77) T3: 0.50 (0.22–0.96)	CRP (mg/L) T1: 3.9 (1.9–7.9) T2: 3.3 (0.7–7.6) T3: 3.8 (1.3–8.9) IL-6 (pg/mL) T1: 2.2 (1.6–3.0) T2: 3.2 (2.3–5.1) T3: 3.1 (1.6–4.8)	Treatment interventions did not significantly change plasma carotenoid levels. However, there was a trend for increased lycopene concentration at visit 2 and 3. Following the interventions, plasma IL-6 and CRP were unchanged.
*Rydén* et al. [71]	Middle-aged men with mild to moderate hypercholesterolemia (*n* = 76)	Placebo (*p*) Simvastatin 40 mg (S)for 6 weeks	*p*: (*n* = 39) S: (*n* = 37)	Plasma Lycopene (nmol/L/cholesterol) Baseline *p*: 116 (89–149) S: 100 (75–142) Week 6 *p*: 125 (98–160) S: 147 (104–182)	CRP (mg/L)/IL-6 (pg/mL) Baseline *p*: 1.1/1.2 S: 1.3/1.5 Week 6 *p*: 1.0/1.3 S: 0.9/1.4	Simvastatin use was associated with significant reductions in CRP and reduced plasma levels of lycopene. However, when adjusted for lipids, lycopene showed significant increases after simvastatin therapy.
*Hurtado-Barroso* et al. [72]	Healthy male subjects (*n* = 22)	Single dose of sofrito (240 g/70 kg)	T1: baseline T2: intervention	Plasma Lycopene (μmol/L) Baseline/After consumption *trans*-lycopene: 2.15 ± 0.30/6.33 ± 1.53 5-*cis*-lycopene: 1.87 ± 0.28/7.93 ± 2.73 13-*cis*-lycopene: 0.21 ± 0.11/2.08 ± 0.78 9-*cis*-lycopene: *n*.d./0.90 ± 0.58	CRP (mg/dL) T1: 0.1 T2: 0.08 IL-6 (pg/mL) T1: 1.4 T2: 1.0TNF-a (pg/mL) T1: 1.0 T2: 0.8	After the sofrito intake, a significant decrease in CRP (*p* = 0.010) and TNF-α (*p* = 0.011) was observed.
*Colmán-Martínez* et al. [73]	Subject at high Cardiovasc. risk (*n* = 28)	Tomato Juice HD 400 mL/d LD 200 mL/d Control: Water for 4 weeks	C: (*n* = 28) LD: (*n* = 28) HD: (*n* = 28)	Plasma Lycopene (μmol/L) *trans*-lycopeneC: 0.70 ± 0.44 LD: 4.04 ± 0.39 HD: 6.67 ± 0.38 5-*cis*-lycopeneC: 1.13 ± 0.28 LD: 2.38 ± 0.27 HD: 4.08 ± 0.26 13-*cis*-lycopeneC: 1.07 ± 0.39 LD: 1.90 ± 0.30 HD: 4.01 ± 0.29 9-*cis*-lycopeneC: 0.42 ± 0.43 LD: 1.05 ± 0.29 HD: 1.92 ± 0.21	CRP (ng/mL) C: 546 ± 46 LD: 442 ± 44 HD: 530 ± 43 IL-8 (pg/mL) C: 40 ± 17 LD: 23 ± 16 HD: 24 ± 15	Plasma lycopene increased significantly in intervention group compared to placebo (*p* < 0.001). No significant alterations in CRP and IL-8 were found.
*Wood* et al. [74]	Asthmatic adults (*n* = 137)	High-antioxidant diet (HAO) or a low-antioxidant diet (LAO) for 14 d Subjects who consumed the low-antioxidant diet received placebo or tomato extract (45 mg lycopene/d).	HAO: (*n* = 46) LAO: (*n* = 91)	Plasma Lycopene (mg/L) Baseline/day 14 HAO: 0.15/0.18 LAO: 0.20/0.13	hsCRP (mg/L)/IL-6 (pg/mL)/TNF-a (pg/L) HAO baseline: 4.2/1.9/1.3 HAO day 14: 3.0/1.9/1.3 LAO baseline: 2.5/1.9/1.4 LAO day 14: 3.3/2.0/1.5	After 14 d of dietary modification, a significant decrease from baseline in plasma lycopene concentrations was observed in the LAO diet group, which was significantly different from the increase in the HAO. No effect of the lycopene-rich supplement compared with placebo was observed. Subjects in the low-antioxidant diet group had increased plasma C-reactive protein at week 14.
*Yeon* et al. [75]	Overweight women (*n* = 22)	High-Vegetable/Fruit (VF) diet (12 servings of VF/day) or low-VF diet (2 servings of VF/day) for 2 weeks, 2 weeks wash-out, 2 weeks	Low base (LB): (*n* = 22) Low post (LP): (*n* = 22) High base (HB): (*n* = 22) High post (HP): (*n* = 22)	Plasma Lycopene (μmol/L) LB: 0.39 ± 0.18 LP: 0.32 ± 0.14 HB: 0.31 ± 0.19 HP: 0.38 ± 0.32	CRP (μg/mL)/IL-6 (pg/mL) LB: 0.54 ± 0.44/3.65 ± 1.51 LP: 0.75 ± 0.70/3.08 ± 0.35 HB: 0.56 ± 0.61/3.52 ± 1.08 HP: 0.40 ± 0.40/3.44 ± 0.83	Results from this study showed that the low-VF diet decreased the average plasma carotenoids by 26%, and the high-VF diet increased the average plasma carotenoids by 32% compared to the baseline values. Changes in plasma lycopene were inversely correlated with changes in plasma IL-6 concentrations when the subjects consumed the low-VF diet.

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
