# Peer review of "The Role of Circulating Lycopene in Low-Grade Chronic Inflammation: A Systematic Review of the Literature"

_molecules, 2020, doi:10.3390/molecules25194378_

Round 1
Reviewer 1 Report
Major comments
This work done on an important and actual heath point has interest but I recommend to make several clarifications in the manuscript before its publication:
1-The authors should better explain why circulating lycopene was selected as a key criteria in this literature review. One could think that beneficial effects of lycopene could follow indirect effects such as: an effect on the microbiota, a digestive action in connection with the intestinal immune system, an increase of the defense capacities against oxidative stress at the hepatic level, ... ) and that therefore circulating lycopene would not be a good criterion to evaluate a beneficial effect.
Also, I recommend to change the title by something like: “The role of circulating lycopene in low-grade chronic inflammation: A systematic review of the literature”.
2- In link with this remark I recommend to add three shorts paragraphs in the introduction on the following points:
2.1- lycopene, its different forms/isomers and physicochemical properties
2.2- lycopene analysis, including a brief discussion of the methods used, to highlight whether analytical determination could be a major biases in the interpretation of results or not;
2.3- kinetic of lycopene after its oral administration: absorption, distribution, metabolism, excretion.
3- Because “Molecular mechanisms of action” (point 4.1 of the manuscript) and “Dietary recommendations” (point 4.2 of the manuscript) are not the purpose of this review I recommend to toggle these paragraphs in the introduction to keep only in the discussion the differences found in the literature cited about circulating lycopene and chronic inflammation.
Detailed comments
L77 : add reference (review) on the use of IL-6 and C-76 reactive protein (CRP) as biomarkers of inflammation
L87-122: A scheme (figure) on the hypothesis of how lycopene can reduce inflammation (may be in link with compounds such as carotenoids or vitamin E) is necessary, specially to justify/explain why circulating lycopene was considered as a key criteria in this review. See also major comments.
L199: check unit: probably µmol/l and not umol/l
L286-288: should be said earlier to a better understanding of the manuscript
L295: yes, and it is why I recommend to do a short paragraph on methods of analysis and kinetic (major comment 2.2 and 2.3)
L299: better to follow if all results were converted to the same unit mg/L or nmol/L but not a mix of two. Use “L” or “l”, but not a mix of two.
L300: check unit
L303-311: yes, and should be said in the introduction. See also major point 2
L313-324: it is why a scheme on hypothesis of the mechanism of action of lycopene is necessary in the introduction
L326-339: same comment, see also comment L87-122
L342: Change “contradictory”. What would have been contradictory would have been to observe a positive correlation in some studies and a negative correlation in others. What you the studies shown is just that the measurement of circulating lycopene is not enough to explain all the cases. To be discussed in relation to the points raised in major comment 2.
L343-363: should be discussed in light to the kinetic of lycopene: rate of absorption, half-life…
L365-368: yes, and it is why a short paragraph on that point is necessary in the introduction (see also major comment 2.1)
L377-466- most of what is said in these lines should be said in the introduction to justify the criteria followed in this review (see major point 3).
L378-390: yes, and it is why saying “conflicting results” is abusive. Like said before, the results of the bibliographic analysis would be easier to follow if hypothesis on the mechanism of action of lycopene have been explained in the introduction to keep only in the discussion of these results in this section on the manuscript.
L391-442; L443-466: this is not a discussion of the results, it would be easier to follow if you keep only in this section what is a discussion about circulating lycopene and chronic inflammation
L468-477: Already said before. It would be preferable to make an opening here on what is not explained by the circulating lycopene by synthesizing the results of the analyzed articles: direct effects on the digestive tract, indirect effect, saturation of the effect due to the ingested dose...
L491-493: remove
Reviewer 2 Report
In this study, the authors performed a systematic literature review to investigate the impact of circulating lycopene on inflammation and the effect of consuming tomato products and/or lycopene supplements on markers of inflammation. Overall it is a well-written manuscript, in which the authors selected 35 published studies. The authors observed from data from 18 selected cross-sectional studies that lycopene levels are adversely affected during inflammation and homeostatic imbalance. Also, they noticed that in most of the 17 selected intervention studies increased circulating lycopene levels after tomato/lycopene supplementation were reported, but almost no changes in inflammation biomarkers were observed. The conclusion of this study was that increasing tomato intake or lycopene supplementation does not diminish inflammation, but depletion of lycopene may be one of the first signs of low-grade inflammation. The article is interesting, adding a new vision on lycopene properties, with an appropriate design and good analysis, but having some minor issues to be corrected.
Specific comments:
1. Some errors occurred when indexing references at page 2, lines 81-85: they are not in increasing order. Please correct this.
2. The authors should provide at least one reference proving that “Inflammatory stimuli, such as IL-6 and C-76 reactive protein (CRP), can then be used as biomarkers to measure inflammation” (page 2, lines 76-77), since this comment was also repeated later in the text.
3. All the fonts in Figure 1 should be increased for a better and easier reading.
4. Tables 1 and 2 need to be improved: (i) all the fonts could be increased, in parallel with a landscape format in page; (ii) individual titles regarding the tables’ content should be provided and (iii) all the references provided do not fit with those mentioned and discussed in the main text, making the parallel reading impossible. Also, at their first mention in the manuscript (page 5, line 172), a small description of the two Tables should be provided.
5. A small error occurred when formatting the Figure 2 legend (page 12, lines 412-420). Also, the fonts could be increased here for a better reading. Another error occurred at page 1, line 10: the line starts with “Faculty”, most probably.
Reviewer 3 Report
The review entitled " The role of lycopene in low-grade chronic 3 inflammation: A systematic review of the literature", by Hidde P. van Steenwijka and collaborators, is an interesting review that collects the scientific literature about the protective the impact of circulating lycopene on inflammation and to investigate the effect of consuming tomato products and/or lycopene supplements on markers of inflammation, and underlying mechanisms of actions in human studies.
In my opinion the review is suitable for publication on this journal after making major revisions.
Also, as a major concern, the review is poorly written in English, with too long sentences, discordance of verbal tenses, mistakes, and sometimes it lacks linearity in entering information and should be subjected to thoroughly editing.
See some examples (not limiting) are listed next:
- Are the articles screened 79 or 80? Review this in the abstract, methods, figure 1 and results.
- Line 120: indicate which are the other biomarkers of inflammation
- Line 230 and line 240: Report data on reference 53 only once in the text. So it's confusing.
- Line 309: specify the relationship between cholesterol and lycopene levels.
- Line 417: compartments????
- Line 421-442: Specify the link between this sentence and the antioxidant effect of lycopene. It gives the impression that some information like curcumin, monoHER .... is out of place!
Round 2
Reviewer 1 Report
Thank you for these changes to your manuscript which I think will improve its readability and interest.
I have one minor comment/suggestion: it would be useful to homogenize the units used to express lycopene contents. This would allow a better comparison of the reported results.
Reviewer 3 Report
The authors responded consistently and correctly to the suggested reviews, the review can be considered for publication in the journal.
Regards